# 3-Iodothyronamine and Derivatives: New Allies Against Metabolic Syndrome?

**DOI:** 10.3390/ijms21062005

**Published:** 2020-03-15

**Authors:** Grazia Rutigliano, Lavinia Bandini, Simona Sestito, Grazia Chiellini

**Affiliations:** Department of Pathology, University of Pisa, 56100 Pisa, Italy; grazia.rutigliano@unipi.it (G.R.); lavinia.bandini@student.unisi.it (L.B.); simona.sestito@for.unipi.it (S.S.)

**Keywords:** metabolic syndrome, 3-iodothyronamine, thyronamine-like analogs, trace amine-associated receptor 1 (TAAR1), obesity, neurodegenerative disorders

## Abstract

In the two decades since its discovery, a large body of evidence has amassed to highlight the potential of 3-iodothyronamine (T1AM) as an antiobesity drug, whose pleiotropic signaling actions profoundly impact energy metabolism. In the present review, we recapitulate the most relevant properties of T1AM, including its structural and functional relationship to thyroid hormone, its endogenous levels, molecular targets, as well as its genomic and non-genomic effects on metabolism elicited in experimental models after exogenous administration. The physiological and pathophysiological relevance of T1AM in the regulation of energy homeostasis and metabolism is also discussed, along with its potential therapeutic applications in metabolic disturbances. Finally, we examine a number of T1AM analogs that have been recently developed with the aim of designing novel pharmacological agents for the treatment of interlinked diseases, such as metabolic and neurodegenerative disorders, as well as additional synthetic tools that can be exploited to further explore T1AM-dependent mechanisms and the physiological roles of trace amine-associated receptor 1 (TAAR1)-mediated effects.

## 1. Introduction

Due to the enormous progress in medical sciences, in the last century Western countries have obtained a conspicuous gain in life expectancy. Nonetheless, these improvements in general health are sensibly undercut by rising rates of overweight and obesity, which currently represent the “leading preventable cause of premature death” worldwide [1]. Overweight and obesity are major determinants of metabolic syndrome, which might be thought of as a silent—people affected are generally asymptomatic—killer, in that it determines increased risk for cardiovascular disease, stroke, diabetes, arthritis, and cancer [2,3,4]. According to the consensus definition agreed upon in 2009, metabolic syndrome is diagnosed when any three of the following five disorders is present: central obesity, insulin resistance, systemic hypertension, high levels of circulating triglycerides and low levels of high-density lipoprotein cholesterol (HDL) [5]. Metabolic syndrome has an alarming prevalence, ranging from about 20% in the Middle East to about 40% in the United States and Europe [6,7,8]. The main triggers for metabolic syndrome are high-calorie diets, consumption of sugary and soft drinks, and physical inactivity [9,10,11,12]. It derives that first-line treatment against metabolic syndrome must be lifestyle changes, with dietary control measures and regular exercise, followed by pharmacological intervention. However, the clinical management of metabolic syndrome needs to be informed by a better understanding of the underlying pathophysiological mechanisms. At present, the most widely accepted hypotheses are insulin resistance with hyperinsulinemia and fatty acid flux, altered inflammatory state, visceral adipose tissue alterations and the activation of the sympathetic nervous system [13,14].

Insulin resistance is thought to play a fundamental role in the development of metabolic syndrome. In the context of high-sugar diets, pancreatic β cells of the Langerhans’ islets will secrete more insulin to maintain blood glucose levels in the normal range. This compensatory mechanism will eventually fail, leading to insulin resistance in the main tissues targeted by insulin, including the adipose tissue, liver, and skeletal muscle. This metabolic downward spiral will finally lead to further inhibition of insulin antilipolytic properties, enhanced free fatty acid flux to tissues, where lipids accumulate due to the positive balance between energy intake and storage capacity [13,15]. Moreover, free fatty acids are toxic to pancreatic β cells, further reducing insulin secretion [16]. Insulin resistance and visceral fat deposits also promote the release of pro-inflammatory cytokines and adipokines, and the activation of neurohormonal pathways. This culminates in the generation of oxygen reactive species (ROS) and inflammation, which contribute to a pro-thrombotic and pro-atherogenic state, with increased risk of cardio-vascular complications [17,18,19].

Thyroid hormone (TH: thyroxine, T4, and triiodothyronine, T3) (Figure 1) crucially controls energy expenditure in the adult. As a matter of fact, hyperthyroid states are characterized by increased resting energy expenditure, weight loss, and heat intolerance, while the opposite—reduced resting energy expenditure, weight gain, and cold intolerance—are hallmark of hypothyroidism [20,21]. While the association between overt thyroid disorders and metabolic syndrome is well established, the effect of thyroid hormone changes on metabolic disturbances in euthyroid subjects or in subclinical hypothyroidism remains unclear. Indeed, depending on study design, setting, population and thyroid status definition, controversial findings have been reported, especially with respect to the validity of mild changes in TSH, T4, and T3 levels within the reference range as risk factors for metabolic syndrome [22].

It is broadly recognized that TH has a “calorigenic effect”, consisting in the maintenance of basal metabolic rate (BMR), and in the facilitation of adaptive thermogenesis. Primarily, TH increases ATP production/consumption, through stimulation of Na^+^/K^+^-ATPase and sarcoplasmic/endoplasmic reticulum Ca^2+^-dependent ATPase (SERCA) toward the potentiation of the respective ion gradients [23,24]. Also, it is hypothesized that TH has an “uncoupling” effect on oxidative phosphorylation in the mitochondria, with a dissipation as heat of the proton-motive force across the mitochondrial inner membrane [25]. Moreover, TH, synergistically with the adrenergic system, is required for facultative thermogenesis in the visceral and subcutaneous brown adipose tissue (BAT), albeit the role of BAT in human metabolism awaits clarification [26]. A minor yet important contribution to BMR is represented by TH-mediated regulation of the intermediary metabolism of carbohydrates, lipids, and proteins, which can be globally conceptualized as an accelerated response to fasting. TH facilitates glucose absorption from the gastro-intestinal tract, and its uptake and oxidation, at the same time dampening glucose-stimulated insulin release, and increasing gluconeogenesis [27]. TH stimulates protein degradation and amino acid release from the skeletal muscle. Particular attention has been devoted to the effects of TH on fat metabolism. In the adipose tissue, TH induces the transcription of enzymes required for fatty acid synthesis, such as acetyl CoA carboxylase (ACC) and fatty acid synthase (FAS), and stimulates lipolysis. Also, TH improves the clearance of cholesterol, and its conversion to bile acids for bile secretion [27]. Recently, it has been suggested that TH metabolic actions might be mediated via central pathways coordinated by hypothalamic nuclei (reviewed in [28]). Neurons in the hypothalamic arcuate nucleus (ARC) receive nutritional and hormonal input, including TH receptor (TR)-mediated input, and send it to the paraventricular nucleus (PVN). In the PVN, information is used—upon integration with signals from other brain regions—to deliver instructions to the periphery, either through the median eminence to the anterior pituitary and endocrine glands, or through the autonomous nervous system (ANS). In addition, TH signalling in the hypothalamus contributes to appetite modulation, by attenuating anorexigenic and enhancing orexigenic pathways, through appropriate changes in the expression of crucial proteins, such as proopiomelanocortin (POMC), uncoupling protein 2 (UCP2), neuropeptide Y (NPY), agouti-related peptide (AgRP), and melanocortin 4 (MC4R) [29].

In summary, TH can promote weight loss and cholesterol reduction, which makes it appealing as a target for developing therapeutic strategies for metabolic disorders. Its application is nonetheless limited by unintended metabolic effects and by those unwanted side effects globally known as thyrotoxic state, consisting of tachycardia, arrhythmia, muscle wasting, tiredness, and osteopenia [30]. Therefore, much effort is currently dedicated to the development of novel TH agonists or analogues with the ability to effectively target metabolic disturbances while being devoid of side effects. After its discovery in 2004, as a putative metabolite of TH, it became rapidly evident that 3-iodothyronamine (T1AM) could exert opposite effects as compared to its precursor TH, including the production of a hypometabolic state and a significant decrease in body temperature. The discovery of this opposite action by T1AM suggested that TH metabolites might also counteract the classical actions mediated by T3, and therefore boosted research on T1AM and related compounds. After almost two decades of investigation, collected evidences point to a far more complex system in which TH and T1AM are either overlapping or antagonistic in the regulation of the metabolism. This review will provide an overview of the most relevant findings contributing to clarify the role of T1AM and related compounds in the regulation of energy homeostasis and metabolism and their therapeutic potential for the treatment of metabolic syndrome.

## 2. Thyronamine: A Novel Pathway of Metabolic Regulation

### 2.1. Endogenous Biosynthesis and Biodistribution of 3-Iodothyronamine (T1AM)

The pioneering finding that 3-iodothyronamine (T1AM), a putative TH derivative (Figure 1), was able to exert remarkable hypothermia and metabolic effects boosted research on this and related compounds. 

Given that T1AM has a β-phenylethylamine structure with one iodine substituent, it has been proposed to derive from TH through decarboxylation and more or less extensive deiodination. At present, the biosynthetic pathway of T1AM remains mysterious. In a rat gut preparation, it was found that T4 or T3 administered exogenously was first deiodinated by type II deiodinase (Dio2) to form T2, followed by ornithine decarboxylase (ODC)-catalysed decarboxylation to 3,5-diiodothyronamine (T2AM), and finally by Dio3-catalysed deiodination to yield T1AM (Figure 2) [31]. Nonetheless, so far these results have not been replicated by other laboratories. The identification of thyronamines (TAMs) as isozyme-specific substrates of deiodinases [32] strongly supports a role of deiodinases in the pathways of TAM biosynthesis. 

While TH is routinely determined via chromatographical or immunological methods, liquid chromatography-tandem mass spectrometry (HPLC-MS/MS) has been initially used for the unequivocal detection of endogenous TAMs. The first report of the endogenous presence of T1AM dates back to 2004, and was obtained using LC-MS/MS [33]. T1AM was found in rodent blood and various tissues (brain, heart, and liver) [33,34]. However, it must be noted that these initial reports were only qualitative in nature. Few years later, a significant technological improvement was achieved with the development of a novel LC-MS/MS method, which allowed the quantitation of T1AM, jointly with TH and/or putative T1AM catabolites in blood and tissue homogenates [35]. This method confirmed that T1AM is indeed an endogenous compound with a widespread distribution. As shown in Table 1, in rodents (i.e., Wistar rat and CD-1 mouse), T1AM concentration in tissues resulted to be significantly higher than in serum (0.3 ± 0.03 pmol/mL), reaching the highest values in rat liver and kidney (92.92 ± 28.46 and 36.08 ± 10.42 pmol/g, respectively). Interestingly, T1AM concentration in tissues revealed to be higher than intracellular T4 and T3 concentration (64.4–141.6 and 0.8–3.072 pmol/mL, respectively) [36,37]. Furthermore, using this method, it was possible to detect T1AM in human blood for the first time. Of note, T1AM serum concentration was found to be very similar between human and rodent serum, namely about 0.15–0.30 pmol/mL [35]. Apparently, contradictory findings were published by Ackermans et al., who were unable detect endogenous T1AM neither in rat/human plasma from rats, nor in thyroid tissue. A possible explanation is that the online solid-phase extraction method (SPE) employed as preanalytical step in their HPLC-MS/MS method could limit the detection of free T1AM [38]. As a matter of fact, the use of different preanalytical extraction methods and HPLC-MS/MS device-dependent limits of detection yielded widely heterogenous measurements of T1AM concentrations, ranging from 0.001 pmol/mL (10^−12^ M; pM) to 1 pmol/mL (10^−9^ M; nM). 

To overcome these analytical problems, a highly specific monoclonal antibody-based T1AM chemiluminescence immunoassay (CLIA) was developed [39]. With this approach T1AM levels in rat tissue and human serum were found in the nanomolar concentration range (14–66 pmol/mL) (Table 1), meaning about two orders of magnitude higher than previously reported in most studies using HPLC-MS/MS [39]. Such huge discrepancy might be related to circulating T1AM being largely bound to lipoprotein particles, particularly to apolipoprotein B100 (apoB100) [42]. It is therefore possible that the typical biogenic amine extraction procedures (for the HPLC method) do not quantitatively liberate T1AM from its binding site within apoB100, and that extraction-based approaches quantify free and not total circulating T1AM, a problem that did not appear with the immunological assay.

### 2.2. Thyronamine Regulation of Metabolism

Since its debut in 2004, it was apparent that T1AM could induce a hypometabolic state, opposite to the effects of its putative precursor, i.e., TH. In that first report, a single intraperitoneal (i.p.) injection of T1AM was able to elicit a rapid (within 30 min) dose-dependent drop in body temperature (from 37 °C to 31 °C at a dose of 50 mg/kg body weight), which was normalized after 6–8 h. Hypothermia was accompanied by profound bradycardia in vivo, and correspondingly, by 63% reduction in cardiac output in an ex vivo working heart preparation [33]. Building on those results, T1AM has received considerable attention in the last decade as a novel modulator of metabolism. The metabolic effect of T1AM was originally tested in the Djungarian hamster *Phodopus sungorus*, a hibernating rodent species, and in mice. Both species, following i.p. injection of T1AM 50 mg/kg body weight, showed a rapid reduction of VO_2_ and respiratory quotient (RQ) from the value of 0.9 to 0.7. The RQ represents an estimate of BMR and is computed as the ratio between bodily carbon dioxide production and oxygen consumption. Since the metabolism of different macronutrients relies on different pathways, the value provides an indication of the energy source used (lipids = 0.7; proteins = 0.8; carbohydrates = 1). Therefore, the authors reasonably speculated that a shift had occurred from prominent carbohydrate to lipid utilization. Hypothermia was slightly delayed after the reduction in metabolic rate, with a strong correlation existing between the two parameters. Such time course suggests that hypothermia might be a consequence of decreased BMR. T1AM appeared to override the ultradian rhythmicity of BMR and body temperature, which became no longer detectable. Also, minor alterations in BMR lasted for more than 24 h. Consistently with a metabolic rerouting toward mobilization and utilization of lipids, a rise in urine ketone body content was observed 8 h after T1AM treatment, while body weight substantially dropped [43]. T1AM has been similarly reported to induce weight loss in several studies [41,44,45,46]. The analysis of body composition revealed that T1AM determined a significant decrease in fat mass, without affecting lean mass [43]. Interestingly, body temperature drop was further worsened by lack of activation of brown adipose tissue thermogenesis and reduced hypomotility, which prevented heat production from muscles [47]. 

It is unclear whether the effects on body weight are associated with changes in food intake, as controversial results have been obtained depending on T1AM dose and route of administration, and in acute vs. (sub) chronic treatments. It should be noted that the dose tested in the aforementioned studies is three orders of magnitude higher than the endogenous levels, casting doubts on the physiological plausibility of the observed biological effects of T1AM. However, in ad libitum fed mice, a single acute i.p. injection of T1AM at much lower doses (1.4 μg/kg) was found to cause a four-fold increase in food intake with non-significant VO_2_ and locomotor activity alterations. The effects lasted for up to 8 h. An even lower T1AM dose was required to elicit a similar effect (40 to 400 ng/kg) upon intracerebroventricular (i.c.v.) or intra-ARC administration. The curve representing the response to increasing T1AM doses was bell-shaped, with the orexigenic effect occurring within a narrow dose range [48]. Partially contradicting results were obtained in fasted mice, where the effect of i.c.v. T1AM on feeding behaviour was biphasic, with a hypophagic response to low doses (1.3 μg/kg), replaced by a hyperphagic response at higher doses (20 μg/kg). The latter was abolished by pre-treatment with the monoamino oxidase (MAO) inhibitor clorgyline, indicating that: (a) The hyperphagic response might actually be mediated by alternative pathways activated by the T1AM oxidative metabolite, the 3-iodoacetic acid (TA1); (b) the increased availability of T1AM may cause a desensitization to orexigenic signals; and (c) the balance between adrenergic drive and the anti-adrenergic effect of T1AM may be moved in favour of adrenergic signalling [49]. 

Evidence regarding the effect of (sub) chronic T1AM treatment of food intake is similarly sparse and inconclusive. A preliminary study reported a reduction in food intake in mice chronically (×14 days) and systemically (i.p.) administered T1AM at high doses (31 mg/kg/day), and that this resulted in weight loss [45]. The effect seemed to vanish at lower T1AM doses (10 mg/kg twice a day × 8 days), which indeed did not change food consumption, despite inducing 8%-body weight loss [46]. 

Recently, the availability of more powerful methods of investigation in the field of obesity—e.g., breath analysis of ^13^CO_2_ by cavity ringdown spectroscopy (CRDS) and nuclear magnetic resonance spectroscopy (NMR)-based metabolomics—allowed to directly demonstrate that T1AM-induced metabolic changes chiefly consist of a shift in macronutrient substrate oxidation from carbohydrates to lipids, as previously postulated. In mice chronically treated with T1AM (10 mg/kg twice a day × 8 days), breath δ^13^C values showed a significant decline in the first 4 days of treatment, pointing to a lipolytic action, which became less apparent from day 5–7 [46]. Blood samples taken from the same mice at the end of the treatment showed: Increased levels of 3-hydroxybutyrate, a lipid catabolism intermediate; increased levels of glycine and valine, products of protein catabolism; and increased levels of acetate. These results indicate that T1AM rapidly induces lipid mobilization, subsequently shifting the fuel source from carbohydrates to lipids, and limiting lipogenesis [46]. Recently, Eskandarzade et al. provided evidence that, similarly to T3, T1AM is also able to enhance browning responses in white adipocytes. Indeed, the authors observed that a chronic low dose of T1AM (10 mg/kg/day; 7 days) increased the levels of UCP1 in mice inguinal white adipose tissue just like T3, and, correspondingly, it was able to manipulate the white adipose tissue for the promotion of thermogenesis and weight loss [50]. 

In a mouse model of polycystic ovary syndrome (PCOS), T1AM administration (25 mg/kg/day × 5 days) induced robust tissue-specific changes in metabolome profiles of plasma, liver and muscle, which were consistent with an inhibitory effect on lipogenic pathways and an increase utilization of fatty acids from lipid degradation as a major energy source in the skeletal muscle [44].

Among the most replicated results, T1AM was found to cause hyperglycaemia and hypoinsulinemia/reduced insulin sensitivity. The first investigations hypothesised that T1AM might act centrally at the hypothalamic level to modify glucose metabolism via sympathetic signalling [28]. Actually, T1AM i.c.v. administration (0.5 mg/kg) in rats did acutely increase endogenous glucose production (EGP), plasma glucose, and glucagon, while insulin tended to decrease [40,51]. Nonetheless, EGP was more marked upon systemic (i.p.) administration [51]. This, together with the observation that changes in glucose metabolism were not affected by pre-treatment with 6-hydroxydopamine, used to produce a transient chemical sympathectomy [52], suggested that T1AM has independent central and peripheral actions. It is tempting to speculate that T1AM can be locally derived from iodothyronines in the central nervous system (CNS).

So far, identified peripheral target tissues of T1AM are the liver and the pancreas. T1AM-mediated hepatic actions were demonstrated in vitro in human hepatocellular carcinoma cells HepG2 and in ex vivo perfused rat liver. In both systems, T1AM had a direct gluconeogenetic effect, that followed a bell-shaped response curve with EC_50_ = 0.8 μM, and was blocked by iproniazid. This was accompanied by concomitant dose-dependent increase in ketone body production, with almost overlapping EC_50_ = 1.1 μM. In brief, in hepatocytes, T1AM is likely to stimulate fatty acid catabolism and shift pyruvate toward gluconeogenesis [53]. In pancreatic β-cells, a single dose of T1AM (50 mg/kg, i.p.) was able to rapidly (within 2 h) induce a 2.5-fold increase in plasma glucose, in parallel with a decrease of insulin and an increase of glucagon levels [52].

### 2.3. Molecular Targets of Non-Genomic Regulation of Metabolism

The rapid development of the metabolic effects of T1AM is inconsistent with the traditional view that TH signaling occurs via gene expression modulation. Indeed, it was soon apparent that T1AM is not a ligand of nuclear TH receptors [33], instead it was found to activate G protein-coupled receptors (GPCRs), in particular trace-amine associated receptor (TAARs). The subfamily of TAARs was discovered as a group of receptors for trace amines (TAM), such as β-phenylethylamine, octopamine, tyramine, and volatile amines, with TAAR1 (trace-amine associated receptor1) being the prototype [54,55,56,57]. TAAR1 has initially been proposed as the main target and effector of TAM action by Scanlan et al. who assayed synthetic iodothyronamines for their ability to stimulate cyclic AMP (cAMP) accumulation in two stable human embryonic kidney (HEK) cell lines. They identified T1AM as the most potent agonist of rat TAAR1. It has also been hypothesized that other TAAR subtypes were targeted by T1AM, such as TAAR5 and TAAR8 [58]. When heterologously expressed, rat, mouse, and human TAAR1 rapidly couple to the stimulation of cAMP production when exposed to T1AM [33]. The high structural conservation of TAAR1 throughout vertebrate evolution highlights the physiological relevance of TAAR1, but also species-specific differences in T1AM potency at TAAR1 orthologs [59]. Upon stimulation, TAAR1 couples to a Gα*_s_* protein and triggers accumulation of intracellular cAMP via adenylyl cyclase activation. This leads to protein kinase A and C (PKA, PKC) phosphorylation and upregulation of transcription factors, namely cAMP response element-binding protein (CREB) and Nuclear factor of activated T-cells (NFAT). In addition, TAAR1 signals via a G-protein independent, β-arrestin2-dependent pathway involving the protein kinase B (AKT)/glycogen synthase kinase 3 (GSK-3) β signaling cascade [60,61].

However, the report that T1AM was able to elicit its characteristic hypothermic effect also in TAAR1-null mice [62] suggests that there must be other molecular targets. Similarly, TAAR1 does not seem to be necessary for the inhibiting effects of T1AM on insulin secretion, which also persist in TAAR1 knockout mice [52]. Indeed, from a theoretical point of view, such inhibitory effect would not be consistent with increased cAMP production at the cellular level. Of note, these experiments were performed in a transgenic mouse model which expresses the catalytic S1 subunit of *Bordetella pertussis* toxin (PTX) in a pancreatic β-cell-specific Cre recombinase-dependent manner. PTX is known to uncouple G_i/o_ proteins from upstream G protein-coupled receptors (GPCRs). As T1AM showed no effect on glucose and insulin levels in RIP-PTX mice, the authors concluded that T1AM negatively modulates insulin secretion in pancreatic β-cells via a G_i_-coupled receptor, which was identified as the α_2A_-adrenergic receptor (ADRA2A) [52]. To further support this hypothesis, T1AM did not cause hyperglycaemia when ADRA2A was absent or blocked by specific antagonists, i.e., yohimbine [52]. Intriguingly, TAAR1 has close structural similarities to other aminergic receptors, especially with adrenergic receptors [63]. Following studies confirmed that T1AM can activate the ADRA2A G_i/o_ signaling pathway almost as efficiently as norepinephrine, indicating a functional role of T1AM at ADRA2A [64].

Such findings seem to be contradicted by the expression of TAAR1 in peripheral organs responsible for food absorption and regulation of glucose homeostasis. TAAR1 mRNA levels were moderate in the stomach (100 copies/ng cDNA), low in the duodenum, and at trace in the pancreas (<15 copies/ng cDNA) [54,55,65]. More in detail, histological analysis of Langerhans islets revealed TAAR1 expression in pancreatic β-cells, but not in glucagon-secreting α cells [66,67]. How to reconcile these apparent contradictions? The response is suggested by the MIN6 insulinoma cell line, which is characterized by a relatively high *Taar1* expression as compared to *Adra2a* [52]. In MIN6 cells, where the pathway initiated at G_s_-coupled TAAR1 dominates, T1AM stimulation increases, rather than decreasing, insulin secretion [52]. On the same line, TAAR1 high-affinity high-specificity ligands, i.e., RO5166017, increased insulin secretion in vitro in INS1E cells. In diabetic mice, this resulted in improved glucose tolerance, increased insulin secretion, reduced plasma glucose, as well as other beneficial effects on glucose homeostasis, such as delaying gastric emptying, stimulating glucagon-like peptide 1 (GLP-1) and peptide YY (PYY), and reducing plasmatic and hepatic triglycerides [67]. In summary, T1AM, at acute high doses, modulates insulin secretion negatively through the interaction with ADRA2A and positively via TAAR1. Whether there exist any pathophysiological conditions in which TAAR1-mediated effects prevail is still unknown. 

So far, it has not been proven that the temperature-reducing effect of T1AM is mediated by GPCRs. Recent studies indicate that this temperature-reducing effect is mediated by peripheral vasodilatation [47]. It was speculated that the anapyrexic effect might be mediated by transient receptor potential (TRP) ion channels [68,69,70,71]. TRPs are Ca^2+^ channels that are widely expressed in several organs, such as hypothalamus, gastrointestinal tract, liver and adipose tissue, where they influence energy homeostasis and thermoregulation [72,73,74,75]. Recently, T1AM was characterized as potent activator of the TRP melastatin 8 (TRPM8) and vanilloid 1 (TRPV1), which are known as cold- and heat-sensitive receptors, respectively [69,70]. Furthermore, the administration of T1AM (10 μM) to hypothalamic cell lines evoked a significant increase of intracellular Ca^2+^ concentration and whole-cell currents, which was strongly inhibited by the TRPM8 selective inhibitor AMTB [76]. 

Notwithstanding T1AM effects on cell membrane proteins, it has been demonstrated that T1AM is taken up by cells, through a specific and saturable mechanism [77]. This mechanism is pH-dependent and does not involve transporters for other monoamines, for organic cations/anions or thyroid hormone. It is possible that these transporters contribute to the regulation of intracellular concentration of T1AM. The presence of a specific transporter suggests the existence of intracellular targets. The first identified intracellular effector of T1AM action in vitro was the mitochondrial F0F1-ATP synthase. Kinetic analyses performed on F0F1-ATP revealed that T1AM, at low, endogenous, concentrations, might affect cell bioenergetics with a positive effect on mitochondrial energy production, by targeting F0F1-ATP synthase [78]. Notably, T1AM was observed to reduce mitochondrial O_2_ consumption and increase H_2_O_2_ release when applied to rat liver mitochondria [79]. In contrast to interactions with receptors at the plasma membrane, T1AM interference with mitochondrial targets occurs at low submicromolar concentrations. Exposing MIN6 cells to T1AM 10 nM resulted in significant reduction in insulin secretion through decreased mitochondrial ATP production [80]. All in all, a complex system is emerging, where metabolic regulation depends on different independent events initiated either at receptors on the plasma membrane or intracellularly upon T1AM uptake (Figure 3).

### 2.4. Genomic Regulation of Metabolism

As mentioned above, some of the metabolic effects of T1AM are long-lasting. This prompted researchers to investigate whether T1AM could modulate gene expression in its target tissues, especially liver, adipose tissue, and skeletal muscle. It was found that T1AM markedly reprogrammed the transcriptional activity of adipose tissue, and, to a lesser extent, of liver (Figure 3). In the adipose tissue, T1AM up-regulated several genes involved in lipoprotein functions (low density lipoprotein receptor adaptor protein 1, *Ldlrap1*; low density lipoprotein receptor-related protein 10, *Lrp10*; Apolipoprotein D, *Apod*; scavenger receptor class B, member 1, *Scarb1*; sirtuin (silent mating type information regulation 2 homolog) 6, *Sirt6*; oxysterol binding protein-like 5, *Osbp15*), with the global effect to promote the clearance of circulating cholesterol and reduce cholesterol biosynthesis. Furthermore, T1AM had regulatory effects on genes related in lipolysis and β-oxidation (alpha(2C)-adrenergic receptor, *Adra2c*; G(0)/G(1) switch gene 2, *G0s2*; acyl-CoA synthetase long-chain family member 5, *Acsl5*; peroxisomal biogenesis factor 5, *Pex5*; Malic Enzyme 1, *Me1*; peroxisome proliferator-activated receptors, *Ppar* α and γ), consistent with stimulation of triglyceride mobilization. Also, the expression of genes related to lipogenesis was influenced by T1AM in the direction of an inhibitory effect on adipocyte differentiation and adipogenesis (CCAAT/enhancer binding protein (C/EBP), beta, *Cebpb*; signal transducer and activator of transcription 5B, *Stat5b*; peripheral myelin protein 22, *Pmp22*; sirtuin [silent mating type information regulation 2 homolog] 2, *Sirt2*; nucleolar and coiled-body phosphoprotein 1, *Nolc1*; insulin-like growth factor binding protein 2, 36kDa, *Igfbp2*; dystrophia myotonica-protein kinase, *Dmpk*; progestin and adipoQ receptor family member III, *Paqr3*; phospholipase A2, group IIA platelets, synovial fluid, *Pla2g2a*) [44,81]. In the liver, the genes whose transcriptional activity was modulated by T1AM were similarly related to different aspects of lipid metabolism, encompassing lipolysis, lipogenesis, and cholesterol uptake (low density lipoprotein receptor adaptor protein 1, *Ldlrap 1*; insulin induced gene 2, *Insig2*; thyroid hormone responsive, *Thrsp*; glycerol kinase, *Gk*; malic enzyme 1, NADP(+)-dependent, cytosolic, *Me1*; D site of albumin promoter (albumin D-box) binding protein, *Dbp*; thyrotrophic embryonic factor, *Tef*; protein tyrosine phosphatase 1B, *Ptp1b*; perilipin 2, *Plin2*; acyl-CoA synthetase long-chain family member 5, *Acsl5*; 3-hydroxy-3-methylglutaryl-CoA reductase, *Hmgcr*) [44,81]. Finally, in the skeletal muscle, T1AM-induced reprogramming of gene expression revealed increased oxidation of fatty acids and their turnover (fatty acid binding protein 3, *Fabp3*; peroxisome proliferator-activated receptor, *Ppar γ*; acyl-CoA synthetase long-chain family member 5, *Acsl5*; situin 4 and 6, *Sirt 4* and *6*) [44] (Figure 3). 

Among proteins whose expression is modulated by T1AM, sirtuins are of particular interest, as they have been recently recognized to be oppositely regulated by metabolic syndrome and calorie restriction [82]. In many organisms, including primates, calorie restriction works to promote survival and protect against disease, such as cardiovascular disease, cancer, diabetes, and neurodegeneration [83,84,85]. Recent findings, which have been brilliantly discussed in [82], suggest that the longevity-promoting effects of calorie restriction might involve sirtuins. These (SIRT1-7) are a class of NAD^+^-dependent deacetylases [86], with an important role in DNA repair and stress response [87]. Furthermore, SIRT1, 3, 4, and 6 can also regulate glucose and fat homeostasis [82], thus representing a possible “entry point” for drugs simultaneously addressing metabolic syndrome and ageing. Intriguingly, T1AM showed (neuro)protective actions in several cell and rodent models, including seizure-related excitotoxic damage, altered autophagy, amyloidosis, and ischemia-reperfusion injury [61,88,89,90,91].

## 3. Pathophysiological Implications

Although the first reported functional—cardiac and hypothermic—effects of T1AM occurred at concentrations three order of magnitude higher than the endogenous levels, it has been later observed that T1AM was able to elicit metabolic actions in relevant tissues at physiologically plausible (submicromolar) dosage. Thus, the physiological and pathophysiological relevance of T1AM in the control of energy homeostasis and metabolism cannot be excluded. However, several issues still await clarification. First, it is still a matter of debate whether T1AM is a derivative of TH in vivo. As mentioned above, T1AM was derived from TH in an intestine preparation by subsequent deiodination and decarboxylation reactions [31]. However, it is unclear whether this is the only biosynthetic pathway or whether T1AM synthesis from T3 may occur in other tissues. In particular, there is evidence that T1AM biosynthesis requires the intact function of the thyroid gland. Upon inhibition of sodium-iodide symporter and thyroperoxidase, used to pharmacologically induce hypothyroidism in rodents, T1AM tissue concentrations were robustly decreased, and, interestingly, remained undetectable even after T4-replacement, despite an almost full recovery of T4 and T3 [92]. Changes in T1AM concentrations might therefore be expected in hypothyroid conditions, and could possibly contribute to some of the symptoms traditionally attributed to TH. Second, the interaction between T1AM- and TH-activated signalling pathways remains to be fully understood. Originally these two systems were thought to be opposite, at least in the heart, where T1AM-mediated negative inotropic and chronotropic effects seemed to work as a brake with respect to those elicited by TH. However, the evidence discussed in this review points to a far more complex system, in which TH and T1AM are either overlapping (e.g., both causing weight loss, reducing insulin secretion and increasing gluconeogenesis) or antagonistic in the regulation of metabolism. Before these findings can be translated to humans, it is mandatory to have a reliable assay for serum T1AM. Despite the current technical shortcomings, preliminary studies in humans have been performed showing that T1AM serum concentrations are correlated to T4 and T3 concentrations [93], and are decreased in non-thyroidal illness syndrome [94]. On the contrary, an increase occurred in type II diabetes [93] and heart failure [95]. In addition, a significant direct correlation was described between serum T1AM concentrations and glycosylated hemoglobulin (HbA1c) and fasting blood sugar [93]. Recently, three rare missense variants of TAAR1 (R23C, S49L, I171L) have been described in a cohort of patients with impaired glycaemia and body weight regulation. The carrier of R23C showed complete loss of insulin production, while the other two carriers were obese/overweight patients with a slight impairment of glucose homeostasis. Accordingly, the functional in vitro characterization of the 3 variants revealed a complete loss of function for R23C, while S49L only partially impaired receptor signaling, and I171L had no effect [96]. Albeit needing confirmation, these findings seem to agree with the hypothesis that there is a balance between T1AM inhibitory and stimulatory effects on glucose-induced insulin secretion in pancreatic β-cells. Derangements from such a balance—either due to increased T1AM or deficient TAAR1-mediated downstream pathway—could lead to altered glucose homeostasis and, eventually, diabetes. 

## 4. Therapeutic Implications and Development Direction

Following the observation that T1AM can induce a hypometabolic state, it was originally thought that antagonizing this pathway could be exploited to raise metabolic rate and facilitate weight loss. Instead, it was later discovered that T1AM would induce long-lasting weight loss without changing food intake, but rapidly converting an animal from carbohydrate to prominent fat oxidation. It thus follows that the selective augmentation of some of T1AM actions can represent an attractive strategy for the treatment of metabolic disturbances [97]. Substantial data described in the last two decades provide compelling evidence of the action of T1AM as a multitarget modulator of metabolism and behavior in several experimental models and pathophysiological conditions [41,44,46,89,90], raising hope for increasing therapeutic option in the treatment of a wide variety of diet- and age-related diseases, such as obesity and neurodegeneration. 

Therefore, one of the biggest challenges for researchers in the coming years will be to conclusively prove whether T1AM might act as a novel “pleiotropic agent” in the context of TH-related diseases, linking endocrine, metabolic and neurodegenerative disorders. On the other hand, given their pleomorphic effects and complex pharmacokinetics, endogenous T1AM and its metabolites may not be the best candidates as novel therapeutic agents. In line with this concept, several research groups have been focusing on the development of synthetic thyronamine-like analogs and/or TAAR1 agonists [98]. In addition to exploit T1AM therapeutic potential, these novel T1AM mimicking agents could also be useful to fully explore the intricate physiological function of T1AM and its metabolites as a whole.

An additional open issue concerns the biosynthetic origin of T1AM. Intriguingly, T1AM was isolated first from rat brain and later shown to be produced in the rat gut from TH through the action of deiodinases and ornithine decarboxylase [31], but the route from gut to the brain still remains an open question, as well as the possible role of gut microbiota in T1AM formation from TH [99], thus encouraging future research in the next years.

## 5. Novel 3-Iodothyronamine Derivatives

In the last decade, even though numerous efforts have been directed to expanding the class of TAM-like compounds, several issues have hampered the discovery of useful new molecules. Taking into account that T1AM interacts with multiple biological targets, including diverse TAARs, namely TAAR1, 2 and 5, one relevant issue in designing TAM-analogs was the selectivity towards a specific target. Furthermore, the limited availability of ligands that preferentially bind hTAAR1 compared to mTAAR1 or rTAAR1 further complicates the search for new species-specific T1AM analogues. An additional concern that needs to be addressed is to avoid the rapid metabolization by enzyme systems such as amino-oxidase (MAO, SSAO), deiodinases (DIO3), sulfotransferases (SULT1A1 and SULT1A3), *N*-acetyltransferases, and glucuronidases, that has been observed for T1AM and could limit the therapeutic application of newly developed molecules [100]. 

The first attempt to develop new T1AM-like molecules was carried out by *Scanlan* and collaborators [101]. This first generation of T1AM analogs featured selected modifications of the molecular scaffold of the endogenous compound, including removal of the phenol hydroxyl group, introduction of a spacer between the two aryl rings, modification in the electronic and steric requirements on the “outer phenyl” moiety, alkylation or modification of the amine, and replacement of the 3-iodo substituent with an alkyl group. Within this new series of T1AM analogues, the benzylated congeners revealed to possess a better TAAR1 affinity [101,102]. Later, our group discovered a novel class of promising thyronamine-like analogs [103,104]. Starting from T1AM, selected structural modifications, including the substitution of the oxygen bridging the two aryl rings with a methylene spacer, replacement of the ethylamine side chain with an aminoethoxy group, replacement of the 3-iodo substituent with hydrogen or methyl group, and bioisosteric transformation of the 4′-OH substituent with 4′-NH_2_, were carried out aiming to (i) obtain more synthetically accessible compounds, and (ii) improve pharmacokinetic properties. These compounds were initially screened in vitro for mTAAR1 activation, and the most promising were evaluated in vivo to characterize their functional and metabolic effects, such as the ability to increase plasma glycaemia and to exert negative inotropic effect. Pharmacological activity on TAAR1 was also rationalized in silico; due to the absence of crystallographic data, a theoretical model of the mTAAR1 receptor was built and used for performing docking studies with the newly developed ligands. Overall, the in vitro and in vivo investigations indicated that derivatives 2 and 3, later named as SG1 and SG2 (Figure 4), were approximately equipotent to endogenous T0AM and T1AM, respectively. 

To validate the use of synthetic T1AM and relative mimickers as therapeutic tools in metabolic diseases, T1AM and SG2 were examined for their ability to modulate lipid metabolism [105]. For this purpose, T1AM and SG2 were administered to 3T3-L1 mouse pre-adipocyte cells either during or after differentiation in mature adipocytes. Results indicated that both compounds were able to induce a significant lipolytic effect, with SG2 being considerably more potent as compared to its natural counterpart. Similarly, T1AM and SG2 reduced total lipid accumulation in HepG2 cells grown under lipogenic conditions. This effect appears to involve the activation of the APMK/ACC signaling, in agreement with the physiological role of this pathway in regulating cellular energy metabolism. 

More recently, Rogowsky et al. investigated the effect of T1AM and its mimickers on cancer cells [106]. Thus, T1AM and SG2 were tested on breast cancer cells (MCF7) and were found to affected cell growth with a potency in the μM range. In particular, SG2 was also effective in reducing cell viability in human liver cancer cells (HepG2), while a far less cytotoxicity was observed in normal cells. Notably, the effect of SG2 on cancer cell viability revealed to be long-lasting, since inhibition of growth rates of cancer cells persisted even after being removed from the media. Although the specific mechanism by which SG2 and T1AM impact cell viability and growth in cancer cells has not been completely elucidated, both compounds are supposed to trigger metabolic disruption of cancer cell function. Particularly, colocalization to the mitochondrial membrane and gene expression analysis hint to an involvement of sirtuin-mediated pathways at the mitochondrial level. 

Moreover, T1AM and its congeners, SG1 and SG2, were subjected to extensive investigations to assess the potential neuroprotection of TAAR1 agonists and the underlying mechanisms [88]. These studies evidenced that systemic administration of T1AM, as previously observed with icv administration, enhances memory and pain sensitivity, two processes strongly impaired in neurodegeneration. Notably, the hyperalgesic effect was lost after pretreatment with the MAO inhibitor clorgyline, which prevents the conversion of T1AM into its oxidative metabolite TA1 (Figure 2). This effect indicates that TA1 might be, directly or indirectly, partially responsible for the pharmacological effects of T1AM, as already observed by Manni and her group [40,49]. Pretreatment with the histamine H1-receptor antagonist pyrilamine triggered a similar effect, supporting the speculation of an involvement of the histaminergic system [107]. Parallel experiments with SG2 and its potential oxidative metabolite, SG6 (Figure 4) produced effects comparable to those of T1AM/TA1, suggesting a common mechanism of action and a good mimic of T1AM functional effects. 

Many evidences indicate an altered autophagic flux as contributing factor in neurodegeneration. Autophagy is essential for neuronal homeostasis and plays a key role in neuronal plasticity. On this basis, the effects induced by T1AM and SG2 in U87MG cell line were explored through ultrastructural investigations and Western blot analysis of common autophagy markers. Both molecules seem to induce autophagy through the inhibition of mTOR phosphorylation by the PI3K/AKT/mTOR pathway [88]. Therefore, the pleiotropic profile outlined for T1AM and its congeners may provide a therapeutic approach also for complex and multifactorial pathologies such as neurodegenerative disorders. 

Overall, SG2 could be considered as the best T1AM-like candidate, useful both as reliable model to design new pharmacotherapeutic agents for metabolic and neurodegenerative diseases [108], and as synthetic tool to further explore T1AM-dependent mechanisms and the physiological roles of TAAR1 receptor.

## 6. Conclusions

T1AM has received considerable attention in the last decade as a novel modulator of metabolism. The results of biomedical studies obtained by several groups provide compelling evidence of the potential of T1AM as a valuable pharmacological tool for the prevention and/or treatment of overweight and obesity, which are the major determinants of metabolic syndrome, one of the major causes of preventable premature death worldwide. Recently, the availability of powerful methods of investigation in the field of metabolomics allowed to directly demonstrate that the chronic treatment of mice with pharmacologic doses of T1AM rapidly induces lipid mobilization, subsequently shifting the fuel source from carbohydrates to lipids, and limiting lipogenesis. A more difficult issue to address has been the identification of the molecular targets responsible for T1AM metabolic regulation. Collectively, all the data available so far suggest that a complex system is involved, where metabolic regulation appear to relay on independent events initiated either at receptors on the plasma membrane, such as TAAR1 and ADRA2A, or intracellularly upon T1AM uptake, including mitochondrial F0F1-ATP synthase. It is well documented that T1AM markedly reprograms the transcriptional activity in its target tissues, especially adipose tissue, liver, and skeletal muscle. Noteworthy, in these metabolically active tissues the genes whose transcriptional activity was modulated by T1AM were similarly related to different aspects of lipid metabolism, encompassing lipolysis, lipogenesis and cholesterol uptake. Among these, sirtuins, a class of NAD^+^-dependent deacetylases, are of particular interest, as they have been recently recognized to play a crucial role in pathways that counter the decline of health that accompanies aging, thus representing a possible “entry point” for pharmacological agents simultaneously addressing metabolic syndrome and ageing. Intriguingly, neuroprotective actions of T1AM have been recently displayed in several human cell lines and rodent models. In order to expand the therapeutic options for the treatment of ageing related disorders, several efforts have been directed to the development of synthetic thyronamine-like analogs and/or TAAR1 agonists. Analog SG-2, which represents a close structural mimic of T1AM, is rapidly emerging as a potent lipolytic and neuroprotective agent. Even though more extensive investigation in animal models of neurodegenerative diseases is still required, SG-2 shows great promise as novel pharmacological agent for the treatment of interlinked diseases, such as metabolic and neurodegenerative disorders.

## Figures and Tables

**Figure 1 ijms-21-02005-f001:**
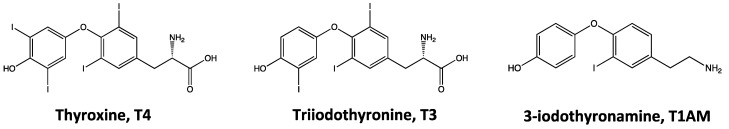
Structures of thyroid hormones (thyroxine, T4, and triiodothyronine, T3) and their putative derivative 3-iodothyronamine (T1AM).

**Figure 2 ijms-21-02005-f002:**
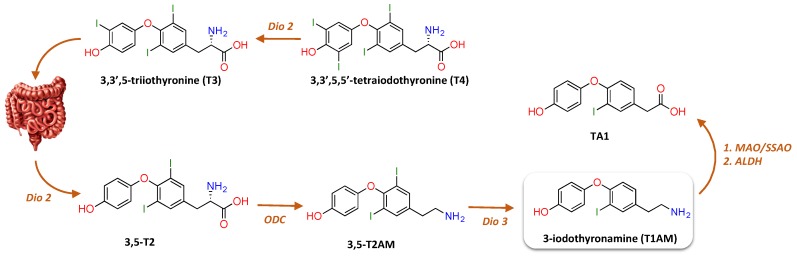
Proposed metabolic pathway to the production of T1AM from T4 in rat gut preparation. As mentioned in the review’s text, in a rat gut preparation it was found that T4 or T3 administered exogenously was first deiodinated by type II deiodinase (Dio2) to form 3,5-T2, which underwent decarboxylation by ornithine decarboxylase (ODC) to generate 3,5-T2AM, followed by Dio3-catalysed deiodination to yield T1AM [31]. Notably, once inside cells, T1AM is rapidly metabolized by ubiquitous enzymes, including monoamine oxidase (MAO), semicarbazide-sensitive amine oxidase (SSAO), and aldehyde dehydrogenase (ADLH), producing 3-iodothyroacetic acid (TA1).

**Figure 3 ijms-21-02005-f003:**
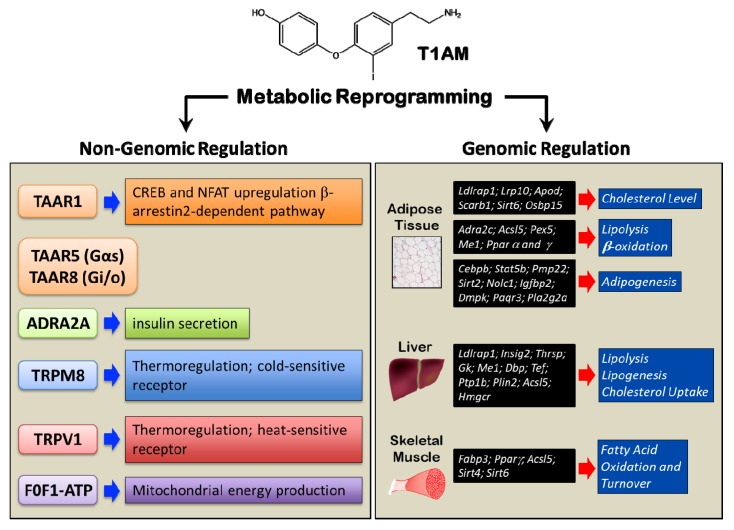
Non-genomic and genomic regulation of metabolism by T1AM.

**Figure 4 ijms-21-02005-f004:**
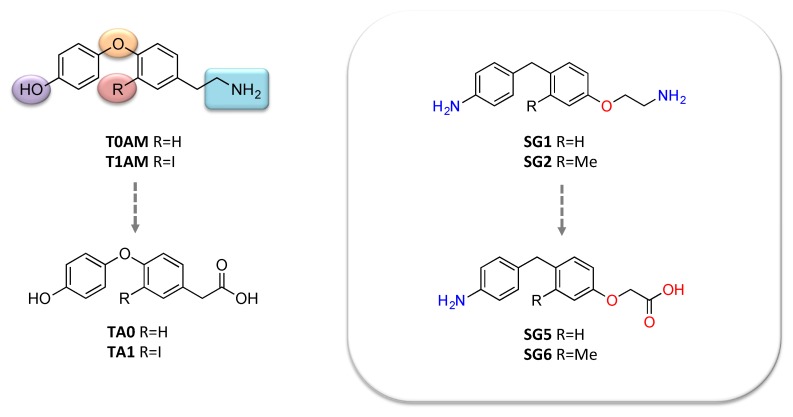
Chemical structures of endogenous thyronamines (T0AM, T1AM), their metabolites (TA0, TA1) and synthetic thyronamine-like derivatives (SG-1, SG-2; SG-5, SG-6).

**Table 1 ijms-21-02005-t001:** 3-Iodothyronamine (T1AM) tissue concentrations in human and rodents.

Compartment		Concentration Range of T1AM	Method	Reference
Human and rat serum		0.15–0.30 pmol/mL	HPLC MS-MS	Saba *et al.* 2010 [35]
		14–66 pmol/mL	CLIA	Hoefig *et al.* 2011 [39]
Rat tissues	Lung	5.61 ± 1,53 pmol/g	HPLC MS-MS	Saba *et al.* 2010 [35]
	Heart	6.60 ± 1.36 pmol/g		
	Stomach	15.46 ± 6.93 pmol/g		
	Muscle	25.02 ± 6.95 pmol/g		
	Kidney	36.08 ± 10.42 pmol/g		
	Liver	92.92 ± 28.46 pmol/g		
Mouse tissues	Brain	0.39 ± 0.102 pmol/g	HPLC MS-MS	Manni *et al.* 2013 [40]
	Liver	7.68 ± 0.85 pmol/g		Assadi-Porter *et al.* 2018 [41]
	Adipose	0.493 ± 0.17 pmol/g		
	Muscle	19.84 ± 3.57 pmol/g		
	Heart	18.15 ± 4.38 pmol/g		

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
