# Peer review of "3-Iodothyronamine and Derivatives: New Allies Against Metabolic Syndrome?"

_ijms, 2020, doi:10.3390/ijms21062005_

Round 1

Reviewer 1 Report

The current review summarized the related researches of 3-iodothyronamine and derivatives against metabolic syndrome. Overall speaking, this review was well organized and easy to be followed. Also, it provided some useful information for the related field. But, I have some comments, please see below.

  1. The current review mainly focused on the 3-iodothyronamine (T1AM). Thus, in my opinion, authors should give more details about T1AM in the part of Introduction, not only clarify the TH.
  2. The pathophysiological mechanisms of metabolic disorders involve many aspects. Why did authors only introduce the high-sugar diet-induced insulin resistance? (Line 44-49) I recommend authors add some contents to establish the link between the paragraphs.
  3. Line 59, “increase” should be revised to “increases”.
  4. Line 98, please give full name for “T2AM”.
  5. For Table 1, could authors add some information about mice? Please revise the format of reference in Table 1.
  6. For Figure 3, error symbol? Please correct it.
  7. T1AM regulates the metabolic signaling of adipose tissue. How about brown adipose tissue? Can you supplement some related researches?
  8. For conclusions, the references should not occur in this section. Please modify it.
  9. Authors should give a single paragraph for describing the current issue and development direction of T1AM-related researches?
  10. Please double-check the style of references, for instance, No. 16, 20…   

Author Response

The current review summarized the related researches of 3-iodothyronamine and derivatives against metabolic syndrome. Overall speaking, this review was well organized and easy to be followed. Also, it provided some useful information for the related field. But, I have some comments, please see below.

We thank Reviewer 1 for the positive feedback.

  1. The current review mainly focused on the 3-iodothyronamine (T1AM). Thus, in my opinion, authors should give more details about T1AM in the part of Introduction, not only clarify the TH.

We agree with the Reviewer, and have added a brief overview of T1AM role in metabolism at the end of the introduction (lines 104-114), as follows:

“After its discovery in 2004, as a putative metabolite of TH, it became rapidly evident that 3-iodothyronamine (T1AM) could exert opposite effects as compared to its precursor TH, including the production of a hypometabolic state and a significant decrease in body temperature. The discovery of this opposite action by T1AM suggested that TH metabolites might also counteract the classical actions mediated by T3, and therefore boosted research on T1AM and related compounds. After almost two decades of investigation, collected evidences point to a far more complex system in which TH and T1AM are either overlapping or antagonistic in the regulation of the metabolism. This review will provide an overview of the most relevant findings contributing to clarify the role of T1AM and related compounds in the regulation of energy homeostasis and metabolism and their therapeutic potential for the treatment of metabolic syndrome.”

  1. The pathophysiological mechanisms of metabolic disorders involve many aspects. Why did authors only introduce the high-sugar diet-induced insulin resistance? (Line 44-49) I recommend authors add some contents to establish the link between the paragraphs.

 A more comprehensive description of the pathophysiological mechanisms involved in metabolic syndrome has been provided, together with four new relevant references (lines 46-57), as follows:

“Insulin resistance is thought to play a fundamental role in the development of metabolic syndrome. In the context of high-sugar diets, pancreatic β cells of the Langerhans’ islets will secrete more insulin to maintain blood glucose levels in the normal range. This compensatory mechanism will eventually fail, leading to insulin resistance in the main tissues targeted by insulin, including the adipose tissue, liver, and skeletal muscle. This metabolic downward spiral will finally lead to further inhibition of insulin antilipolytic properties, enhanced free fatty acid flux to tissues, where lipids accumulate due to the positive balance between energy intake and storage capacity [13,15]. Moreover, free fatty acids are toxic to pancreatic β cells, further reducing insulin secretion [16]. Insulin resistance and visceral fat deposits also promote the release of pro-inflammatory cytokines and adipokines, and the activation of neurohormonal pathways. This culminates in the generation of oxygen reactive species (ROS) and inflammation, which contribute to a pro-thrombotic and pro-atherogenic state, with increased risk of cardio-vascular complications [17-19].”

  1. Tooke, J.E.; Hannemann, M.M. Adverse endothelial function and the insulin resistance syndrome. J Intern Med 2000, 247, 425-431, doi:10.1046/j.1365-2796.2000.00671.x.
  2. Juhan-Vague, I.; Alessi, M.C.; Mavri, A.; Morange, P.E. Plasminogen activator inhibitor-1, inflammation, obesity, insulin resistance and vascular risk. J Thromb Haemost 2003, 1, 1575-1579, doi:10.1046/j.1538-7836.2003.00279.x.
  3. Wallace, A.M.; McMahon, A.D.; Packard, C.J.; Kelly, A.; Shepherd, J.; Gaw, A.; Sattar, N. Plasma leptin and the risk of cardiovascular disease in the west of Scotland coronary prevention study (WOSCOPS). Circulation 2001, 104, 3052-3056, doi:10.1161/hc5001.101061.
  4. Mehta, P.K.; Griendling, K.K. Angiotensin II cell signaling: physiological and pathological effects in the cardiovascular system. Am J Physiol Cell Physiol 2007, 292, C82-97, doi:10.1152/ajpcell.00287.2006.

We hope this improves the clarity of the text.

  1. Line 59, “increase” should be revised to “increases”.

Ok. Thank you.

  1. Line 98, please give full name for “T2AM”.

 We thank the Reviewer for spotting the error. We have now provided the full name for T2AM, i.e., 3,5-diiodothyronamine.

  1. For Table 1, could authors add some information about mice? Please revise the format of reference in Table 1.

 As requested, in Table 1 we have added information about T1AM concentration in mouse tissues. The format citation has been revised.

  1. For Figure 3, error symbol? Please correct it.

 We thank the Reviewer for detecting the error symbol, it has now been corrected.

  1. T1AM regulates the metabolic signaling of adipose tissue. How about brown adipose tissue? Can you supplement some related researches?

 We thank the Reviewer for pointing to the interesting topic of brown adipose tissue. Due to the effects of T1AM on body temperature, it is tempting to speculate that a modulation of brown adipose tissue thermogenesis might be involved in T1AM metabolic actions. However, evidence about the effects of T1AM on brown adipose tissue is scarce. We have now supplemented our manuscript with the observations of Gachkar, et al. (2017), who found that T1AM-induced hypothermia was not counteracted by activation of brown adipose tissue (lines 196-198), as follows:

“Interestingly, body temperature drop was further worsened by lack of activation of brown adipose tissue thermogenesis and reduced hypomotility, which prevented heat production from muscles [46]”

In addition, more related to T1AM-activated metabolic signalling in adipose tissue, it has been recently found that T1AM promotes a “browning” of white adipose tissue. These results have been reported in lines 232-237, as follows:

“Recently, Eskandarzade et al. provided evidence that, similarly to T3, T1AM is also able to enhance browning responses in white adipocytes. Indeed, the authors observed that a chronic low dose of T1AM (10 mg/kKg/day; 7 days) increased the levels of UCP1 in mice inguinal white adipose tissue (IWAT) just like T3, and, correspondingly, it was able to manipulate white adipose tissue for promotion of thermogenesis and weight loss [49].”

  1. For conclusions, the references should not occur in this section. Please modify it.

We have removed all references from the conclusion section, as suggested.

  1. Authors should give a single paragraph for describing the current issue and development direction of T1AM-related researches?

We appreciate the Reviewer’s suggestion. We expand the paragraph entitled “Therapeutic implications” by adding “development direction”.

“Substantial data described in the last two decades provide compelling evidence of the action of T1AM as a multitarget modulator of metabolism and behavior in several experimental models and pathophysiological conditions [42, 43, 45, 89, 90], raising hope for increasing therapeutic option in the treatment of a wide variety of diet- and age-related diseases, such as obesity and neurodegeneration.

Therefore, one of the biggest challenges for researchers in the coming years will be to conclusively prove whether T1AM might act as a novel “pleiotropic agent” in the context of TH-related diseases, linking endocrine, metabolic and neurodegenerative disorders. On the other hand, given their pleomorphic effects and complex pharmacokinetics, endogenous T1AM and its metabolites may not be the best candidates as novel therapeutic agents. In line with this concept, several research groups have been focusing on the development of synthetic thyronamine-like analogs and/or TAAR1 agonists [98]. In addition to exploit T1AM therapeutic potential, these novel T1AM mimicking agents could also be useful to fully explore the intricate physiological function of T1AM and its metabolites as a whole.

An additional open issue concerns the biosynthetic origin of T1AM. Intriguingly, T1AM was isolated first from rat brain and later shown to be produced in the rat gut from TH through the action of deiodinases and ornithine decarboxylase [31], but the route from gut to the brain still remains an open question, as well as the possible role of gut microbiota in T1AM formation from TH [99], thus encouraging future research in the next years.”

  1. Please double-check the style of references, for instance, No. 16, 20

 The style of the reference list has been carefully double-checked.

Please also see the attachment.

Reviewer 2 Report

In this article, Rutigliano et al deeply revised the effect of 3-iodothyronamine and derivatives as regulator of metabolism. The review is overall well done, and I have very few comments, reported hereunder:

Definition of metabolic syndrome reported in lines 32-34 is not correct. Metabolic syndrome is defined as the presence of 3 out 5 of its components (central obesity, high blood pressure, hyperglycemia, high triglycerides, and low HDL). I would suggest to add a short paragraph at the end of introduction to underline that despite the great body of evidences reported by the authors, the real net effect on human is not clear. Indeed, studies on the association between disorders of the thyroid gland and metabolic syndrome are not univocal as reported in the following review (doi: 10.2174/1871530317666170320105221) that should be cited in the text.  A recent review on the role of thyroid ormone integration of metabolic pathways in the central regulation of metabolism has been published and I suggest to cite in the text DOI: 10.3390/ijms19072017.

Author Response

In this article, Rutigliano et al deeply revised the effect of 3-iodothyronamine and derivatives as regulator of metabolism. The review is overall well done, and I have very few comments, reported hereunder

We thank Reviewer 2 for the positive feedback.

1. Definition of metabolic syndrome reported in lines 32-34 is not correct. Metabolic syndrome is defined as the presence of 3 out 5 of its components (central obesity, high blood pressure, hyperglycemia, high triglycerides, and low HDL).

We thank the Reviewer for his/her correction. We have modified the definition
(lines 33-36), as follows:

“According to the consensus definition agreed upon in 2009, metabolic syndrome
is diagnosed when any three of the following five disorders is present: central
obesity, insulin resistance, systemic hypertension, high levels of circulating
triglycerides and low levels of high-density lipoprotein cholesterol (HDL) [5].”

2. I would suggest to add a short paragraph at the end of introduction to underline that despite the great body of evidences reported by the authors, the real net effect on human is not clear. Indeed, studies on the association between disorders of the thyroid gland and metabolic syndrome are not univocal as reported in the following review (doi: 10.2174/1871530317666170320105221) that should be cited in the text.

We thank the Reviewer for highlighting the important issue of the role of mild
changes in thyroid hormone levels – such as in euthyroid subjects or in subclinical hypothyroidism – in metabolic syndrome. As the Reviewer indicated, this topic is still open to debate, as no conclusive results have been obtained so far. We added the following short paragraph in the introduction (lines 62-67), with reference to: Delitala, A.P.; Fanciulli, G.; Pes, G.M.; Maioli, M.; Delitala, G.
Thyroid Hormones, Metabolic Syndrome and Its Components. Endocr Metab
Immune Disord Drug Targets 2017, 17, 56-62.

“While the association between overt thyroid disorders and metabolic syndrome is well established, the effect of thyroid hormone changes on metabolic
disturbances in euthyroid subjects or in subclinical hypothyroidism remains unclear. Indeed, depending on study design, setting, population and thyroid status definition, controversial findings have been reported, especially with respect to the validity of mild changes in TSH, T4 and T3 levels within the reference range as risk factors for metabolic syndrome [22].”

3. A recent review on the role of thyroid hormone integration of metabolic pathways in the central regulation of metabolism has been published and I suggest to cite in the text DOI: 10.3390/ijms19072017.

Following the Reviewer’s suggestion, we have discussed the central effects of
thyroid hormone on appetite, citing the recommended reading (lines 94-98), as
follows:

“In addition, TH signalling in the hypothalamus contributes to appetite
modulation, by attenuating anorexigenic and enhancing orexigenic pathways,
through appropriate changes in the expression of crucial proteins, such as
proopiomelanocortin (POMC), uncoupling protein 2 (UCP2), neuropeptide Y
(NPY), agouti-related peptide (AgRP), and melanocortin 4 (MC4R) [29].”

Kouidhi, S.; Clerget-Froidevaux, M.S. Integrating Thyroid Hormone Signaling in
Hypothalamic Control of Metabolism: Crosstalk Between Nuclear Receptors. Int
J Mol Sci 2018, 19, doi:10.3390/ijms19072017.

Please also see the attachment.

Round 2

Reviewer 1 Report

Thank you for your response. The quality of current manuscript has been improved. I have no further comments.